# The Influence of Self-Expansion and Consumer Engagement on Consumers’ Continuous Participation in Virtual Corporate Social Responsibility Co-Creation

**DOI:** 10.3390/bs13070545

**Published:** 2023-06-29

**Authors:** Jinjun Nie, Xiaoyi Wang, Chan Yang

**Affiliations:** 1Qianjiang College, Hangzhou Normal University, Hangzhou 310018, China; njj2021@huqc.edu.cn; 2School of Management, Zhejiang University, Hangzhou 310012, China; 3Neuromanagement Lab, Zhejiang University, Hangzhou 310012, China; 4College of Economics and Management, Zhejiang Normal University, Jinhua 321004, China; yangchan@zjnu.cn

**Keywords:** virtual CSR co-creation, self-expansion, consumer engagement, continuous participation

## Abstract

Virtual corporate social responsibility co-creation (VCSRC) became an effective strategic tool with which enterprises can fulfill social responsibilities and retain customers. This study investigated the drivers of consumers’ continuous participation in VCSRC based on online survey data collected from 336 VCSRC participants. From a new perspective of self-expansion theory and by integrating consumer engagement (CE), we constructed a theoretical model and proposed a set of hypotheses, which were tested by using the structural equation model (SEM). Our findings show that self-expansion (experience-based expansion, competence-based expansion, and identity-based expansion) has a significant positive impact on continuous participation, with CE (conscious attention, enthusiasm, and social connection) playing a partial mediating role. Our research not only theoretically contributes to the research on VCSRC and self-expansion theory, but also inspires the operation of VCSRC projects in motivating consumers’ continuous participation.

## 1. Introduction

The emergence of social media increasingly integrated stakeholders in companies’ corporate social responsibility (CSR) processes called virtual CSR co-creation (VCSRC), which is defined as a company utilizing social media tools to engage stakeholders in CSR activities [1,2]. Examples of such practices include Future Friendly Challenge (Procter & Gamble), Sustainable Living Lab 24 h (Unilever), Ideas Brewery (Heineken), and Refresh Everything (Pepsi). The most representative practice in China is Ant Forest, which is a virtual platform aimed at environmental protection launched by Alibaba in 2016. By August 2022, Ant Forest drove 650 million people to practice green consumption, planted more than 400 million trees, and achieved 26 million tons of carbon emission reduction. Some virtual CSR co-creation projects similar to Ant Forest achieved sustainable development and reaped good social effects, while others are gradually declining [3]. The key to success lies in driving consumers’ continuous participation and establishing long-term interaction relationships with consumers [2]. As a result, how to promote consumers’ continuous participation is an urgent problem for the sustainable development of VCSRC projects, and understanding what motivates such participation is an issue of major significance, yet few studies addressed this topic.

Consumers’ participation in VCSRC refers to various behaviors, including the dedication of resources (time, vigor, knowledge, money, networks, emotion, etc.), interpersonal interaction, and the practice of responsible behaviors via the virtual platforms [4]. Following Osei-Frimpong et al. [5], we define continuous participation in VCSRC as consumers’ repetitive participation in VCSRC for at least six months. Previous studies used various theories to investigate users’ continuous participation in a social media context, such as the expectation confirmation theory (e.g., [6,7]), theory of planned behavior (e.g., [8]), technology acceptance model (e.g., [9,10]), and uses and gratifications theory (e.g., [5,11]), generally from the perspectives of hedonism or utilitarianism. However, consumers’ participation in VCSRC is entirely voluntary because it fulfills the social responsibility of contributing to public welfare efforts [12]. It is a socially responsible behavior with altruistic attributes and intrinsic moral factors [13], as well as a virtual co-creation activity generally designed with incentive gamification (using game designs in non-game contexts [14]) principles. Therefore, the theoretical perspectives of hedonism or utilitarianism may not be sufficient to explain the emergence of VCSRC [15]. Just as Zhang et al. [16] found, utilitarian value does not have a positive effect on user’s attitude to Ant Forest. Therefore, we adopted the self-expansion theory to examine consumers’ continuous participation in VCSRC.

Self-expansion is a fundamental motive to increase one’s self-concept through the acquisition of resources, perspectives, and identities that enhance one’s ability to accomplish goals [17]. It can be achieved by engaging in novel, exciting, and interesting activities, or including the qualities of one’s close friends and family [18,19]. The self-expansion theory contends that people are motivated to expand the self, to know or become more, to move closer to a state of wholeness, and to expand their sense of place in the universe [20]. Previous research showed that self-expansion might correlate with inner moral justice in the face of social responsibility [13]. In addition, self-expansion is a psychologically healthy motivation that contributes to mental health, well-being, and a “broadened identity or awareness, which usually leads to greater altruism” [13,20]. Moreover, self-expansion plays an important role in building and maintaining relationships [21]. Self-expansion is proven to be a significant factor affecting loyalty in intimacy [18]. In recent years, research on self-expansion extended to non-relational contexts, such as consumption [22], travelling [23], mobile phone usage [24], participation in communities [25], pet ownership [26], and internet usage [27]. The positive impact of self-expansion on loyalty and continuation in non-relational contexts was also noted. For instance, Liu et al. [28] proved that self-expansion positively affects user loyalty in smartwatch usage. However, there is currently little research on the influence of self-expansion on loyalty and continuation in non-relational contexts, which urgently needs to be explored. For these reasons, using self-expansion theory to explore consumers’ continuous participation in VCSRC is a suitable aspect for study. We argue that VCSRC platforms, with their capability to offer multi-dimensional experiences and resources, can trigger consumers’ self-expansion to promote relationship development with the platforms.

A few studies contributed to this issue of consumers’ continuous participation in VCSRC from perspectives beyond hedonism or utilitarianism. For example, Ashfaq et al. [29] suggested that altruism and self-promotion have positive impacts on Ant Forest users’ intention for continued participation; Zhang et al. [16] demonstrated that self-transcendence values positively affect users’ attitude toward Ant Forest. However, the existing studies lack examination of the influence mechanism of the motivational factors on consumers’ continuous participation, and we therefore lack in-depth knowledge of these drivers of consumers’ continuous participation behavior in VCSRC. To fill this gap, we introduce consumer engagement (CE) as a mediating factor of the relationships between self-expansion and continuous participation to investigate the influence mechanism.

In this study, CE is defined as a consumer’s psychological state with respect to their co-creating experience with the platforms, enterprises, and other consumers. It was proven that CE contributes to continuance intention [30,31,32,33] and customer loyalty [34,35,36]. The high degree of CE determines the sustainable development of virtual platforms [37], and the VCSRC platforms are no exception. Moreover, CE is the micro-foundation of value co-creation [38], while the essence of virtual CSR co-creation is value co-creation [1,2]. Therefore, we considered CE as an important factor for consumers’ continuous participation in VCSRC. However, although the positive effects of CE on continuous participation were confirmed in some studies, there is no consensus on the definition and dimensionality of CE in these studies. Further exploration is needed to understand the connotation and dimensionality of CE in the VCSRC context.

Previous studies showed that consumers’ psychological factors are key antecedents for CE, such as trust [35], entertainment [32], goal pursuit [39], emotional attachment [39], social presence [32], and self-identification [40]. However, whether the motivation of self-expansion is the antecedent of CE is yet to receive enough attention, and whether CE is a conduit between self-expansion and continuous participation in VCSRC situations remains unclear and warrants further elucidation. Moreover, the existing studies mostly treated self-expansion as a single dimensional concept and rarely considered the subdivision of dimensions, which is also the concern of this study. Thus, this study investigated the detailed relationships among self-expansion, consumer engagement, and consumers’ continuous participation in VCSRC. We contend that the motivation of self-expansion drives consumers to engage with VCSRC platforms, further encouraging consumers to maintain a reciprocal relationship with the platforms.

This paper is structured as follows: Section 2 contains a literature review. Section 3 outlines the research hypotheses and the theoretical model. Section 4 presents the methods. Section 5 shows the results of the empirical study. Section 6 presents the conclusion and discussion of the research.

## 2. Literature Review

### 2.1. Consumers’ Participation in Virtual CSR Co-Creation

The concept of CSR co-creation was first proposed by Morsing et al. [41], and was defined as the development of CSR activities on virtual platforms, enabling stakeholders to design and take part in CSR activities. It was a form of social value co-creation based on virtual CSR activities in which enterprises communicate and collaborate with stakeholders through social media platforms [1,2]. Consumers participating in CSR co-creation is a significant way to fulfill social responsibility for the benefits of others or society [42], including positive extra behaviors such as customer resource dedication (time, vigor, money, knowledge, networks, emotions, etc.), interpersonal interaction, and social responsibility behaviors [42,43,44,45]. Continuous participation in this study is conceptualized as the repetitive participation in VCSRC through social media platforms, which is not limited to the frequency of accessing the platform, but includes how long a person is using the platform and following the CSR co-creation activities.

Some driving factors of consumers’ continuous participation in VCSRC attracted the attention of a few scholars, such as consumer factors, virtual platform factors (e.g., game design elements [46]), enterprise factors (e.g., corporate reputation [47]), convenience [48], and environmental factors (e.g., environmental concern [46]). Among these factors, consumer factors received considerable attention. Consumers’ values [16,48], a sense of achievement and perceived entertainment [12], personal attributes [29], social motivation [49], gratification [11], self-promotion [29], and self-transcendence [16] were proven to have positive impacts on continuous participation. The existing studies either focus on hedonic or utilitarian perspectives, or lack examination of the influence mechanism of motivational factors on consumers’ continuous participation, which cannot fully explain consumers’ continuous participation in VCSRC, a socially responsible behavior with altruistic attributes and intrinsic moral factors [13]. Further exploration of the driving factors is needed.

### 2.2. Self-Expansion Theory

Self-expansion is a motive to increase one’s self-concept by adding positive content, such as novel resources, perspectives, and identities, to the self-concept [17]. The positive content can be obtained by partaking in novel, exciting, and interesting activities, or by adopting the qualities of one’s close friends and family [18,19]. The self-expansion theory was initially applied in the field of intimate relationships. With the deepening of research, it was applied to a broader range of topics. Aron et al. [18] proposed two ways of achieving self-expansion: completing new and challenging tasks that can enhance one’s self-worth and incorporating other people or entities into oneself to directly gain new perspectives, identities, and abilities. Studies proved that individuals include varied entities in the self, such as products and brands [50,51], mobile phones [24], pets [26], community [25], music festivals [52], and the internet [27]. Self-expansion can occur in various contexts, such as meeting new people, travelling, trying new hobbies, visiting museums, and learning new information [23]. These studies provided evidence that other people and entities can also be the source of self-expansion in addition to intimate relationships. Table 1 lists some self-expansion-related studies.

Self-expansion was proven to be a significant factor affecting loyalty in intimacy [18]. Recently, the positive impact of self-expansion on loyalty and continuation in non-relational contexts was also noted. For instance, Liu et al. [28] proved that self-expansion positively affects user loyalty in smartwatch usage. By expanding the self, individuals might feel an enhancement in their abilities and have more resources to make use of, thereby increasing feelings of self-efficacy, motivating them to put more efforts into subsequent tasks [53,54], and leading to positive outcomes in terms of relationships, health, and well-being [55]. In this study, VCSRC platforms can provide rich resources and activities to facilitate consumers’ goal achievement. We argue that self-expansion theory makes it possible to introduce the idea that the resources provided by the VCSRC platforms can lead them to be included in the consumers’ self-concept and, consequently, encourage engagement and facilitate continuous participation. Hence, incorporating self-expansion theory in this study to reveal consumers’ continuous participation behavior is considered suitable.

**Table 1 behavsci-13-00545-t001:** Self-expansion-related studies.

Reference	Study Focus	Methods	Main Findings
Lewandowski and Ackerman (2006) [56]	Self-expansion as predictors of susceptibility to infidelity	Survey study Sample size: 109 participants	When a relationship was not able to fulfill needs or provide ample self-expansion for an individual, their susceptibility to infidelity increased.
Mattingly and Lewandowski (2013) [57]	Benefits of individual self-expansion	Experimental study	Individuals who engaged in high self-expanding activities exerted more effort in cognitive and physical tasks than those who engaged in low self-expanding activities.
Mclntyre et al., (2014) [58]	Workplace self-expansion	Survey study Sample size: Study 1:84 Study 2:73	Greater workplace self-expansion was associated with greater job satisfaction and commitment and leaving a self-expanding job was associated with negative consequences for the self.
Shedlosky-Shoemaker et al., (2014) [59]	Self-expansion through fictional characters	Survey study Sample size: 113 participants	Immersion into a narrative led to greater cognitive overlap with the character and perceived self-expansion.
Hoffner et al., (2015) [24]	Self-expansion via mobile phones	Survey study Sample size: 272 participants	Self-expansion via mobile phones was associated with greater inclusion of the mobile phone in the self-concept and greater subjective well-being.
Branand et al., (2015) [25]	Inclusion of college community in the self-concept	4-year, six-wave longitudinal study of one cohort of college students	Inclusion of the college community in the self was a central, mediating link between involvement and satisfaction.
Slotter and Kolarova (2019) [60]	Self-esteem and spontaneous self-expansion	Experimental study	When presented with a prospective romantic partner, higher self-esteem people would self-expand to adopt positive attributes, while lower self-esteem people would self-expand to adopt negative attributes.
Lee et al., (2019) [61]	Pop star fans’ self-expansion that affects their travel behaviors	Survey study Sample size: 219 participants	Pop star fans’ self-expansion was a significant motivation to seek fan club membership and led to a positive relationship with their favorite pop star’s country.
Harasymchuk et al., (2020) [23]	Antecedents of relational self-expansion	Survey study Sample size: 122 participants	Daily goals, particularly goals oriented toward achieving positive relationship outcomes, were associated with a greater likelihood of engaging in exciting activities occurring with a partner, which, in turn, led to higher daily relationship self-expansion.
Gorlier and Michel (2020) [51]	Self-expansion in consumer–brand relationships	Experimental study	Compared to mundane rewards, special rewards produced higher self-expansion, which in turn led to positive brand evaluation.
Michel et al., (2022) [22]	Self-expansion in consumer–brand relationships	Survey study Sample size: 2010 participants	Brands can generate self-expansion through the embodiment of personal ideals, which in turn leads to more favorable consumer responses to the brand.
Apaolaza et al., (2022) [26]	Self-expansion through pets	Survey study Sample size: 326 participants	Self-expansion moderated the effect of pet anthropomorphism on the purchasing of fashion pet clothing.
Niu et al., (2023) [27]	Internet self-expansion	Survey study Sample: Four groups of participants	A three-factor model with 16 items was developed and tested to measure internet self-expansion.
Liu et al., (2022) [28]	Self-expansion through smartwatch usage	Survey study Sample size: 343 participants	Smartwatch use positively affected self-expansion and self-expansion positively affected user loyalty and user influence.

### 2.3. Consumer Engagement through Social Media

CE obtained considerable academic and practical attention in recent years [62,63]. However, there is not yet a consensus on the definition of CE. Previous studies mainly defined CE from three perspectives, including behavior (e.g., [44,64]), psychological state (e.g., [65,66,67]), and psychological process (e.g., [68]). The psychological state perspective is the most widely accepted. CE is generally viewed as a multi-dimensional construct covering cognitive, emotional, and behavioral components [67,69,70,71,72,73]. Some scholars expanded the research framework to incorporate the social elements of CE [65,67], which entails socializing with others in virtual communities [72,73]. For example, Vivek et al. [70] proposed a CE scale including three dimensions of “conscious attention, enthused participation, and social connection.”

The rise of social media offers firms good platforms to engage with consumers. Much of the research on CE focused on online and social media contexts such as social media brand engagement (SMBE) [5,65]. Recent reviews shed light on the contextual characteristics for assessing CE [74]. The object of CE may vary depending on the context of the research, which might be brand, service product, game, mobile application, community, media, activities, events, etc. [75,76,77]. The variety of CE contexts leads to different ways to define CE. Regarding our study object, VCSRC platforms are mostly designed with gamified elements, including consumer rewards and medals, to attract consumers, motivate them to perform planned behaviors, and achieve meaningful objectives. Highly engaged consumers with strong enthusiasm can provide VCSRC platforms an active online environment and ensure the effectiveness of the co-creation activities. On this basis, this study defined consumer engagement as a consumer’s psychological state consisting of conscious attention, enthusiasm, and social connection dimensions with respect to their co-creating experience with the platforms, enterprises, and other consumers. Conscious attention refers to a person’s degree of interest in VCSRC platforms; enthusiasm refers to the zealous feelings and reactions of a person related to using the platforms; social connection refers to the enhancement of interactions with others on the platforms, suggesting mutual or reciprocal behaviors [77].

Past research highlighted that information [78], entertainment [79], social interaction [80], personal identity [81], goal pursuit [39], altruism [81], self-image expression [78], remuneration [82], and hedonic and utilitarian motives [32] serve as antecedent factors for CE in the social media context. Consumers’ needs, motives, and goals invoke various psychological states, which in turn drive their participation [14]. Motivation has multiple aspects. It is believed that when consumers are internally motivated, their engagement degree is significantly higher than that from external incentives [83,84]. Advancing this stream of CE research, this study identified self-expansion as another internal motivational driver of CE.

There is a consensus that CE is important for the success of marketing and sales activities [85,86]. Studies proved that CE is a factor of great importance in predicting customer loyalty or continuance [33,35,65,87,88]. In the social media context, customer loyalty is mainly manifested in repeated visits [87], continuous use [40,67], and positive word-of-mouth recommendations [89,90]. Moreover, scholars agree that CE and value co-creation are inextricably linked [66,91]. Within virtual contexts, the CE concept in particular provides a useful way to understand the value co-creation of enterprises and consumers [38].

## 3. Hypotheses Development and Conceptual Model

### 3.1. Self-Expansion and Consumers’ Continuous Participation

Self-expansion can be achieved by acquiring new perspectives, resources, knowledge, insights, abilities, and identities through partaking in novel, interesting, exciting activities, etc., that lead to increased self-efficacy and an enhanced sense of self [51,56]. Consumers’ participation in VCSRC is a challenging and creative activity and a process of continuous self-transcendence and pleasant experiences [92]. VCSRC platforms, similar to brands, can provide consumers with a wealth of resources to help them achieve self-expansion and enhance their abilities to achieve their goals. Drawing on Niu et al. [27], we divided self-expansion into three dimensions (experience-based expansion, competence-based expansion, and identity-based expansion) and believe that these three dimensions all have significant positive impacts on consumers’ continuous participation in VCSRC.

Experience-based expansion is the acquisition of new experiences and perspectives. VCSRC platforms provide new experiences to consumers by establishing convenient, effective ways for them to participate in social responsibility activities. Moreover, VCSRC platforms widely adopted game design elements that evoke pleasant experience and open up new perspectives, including meaningful media experience, consumer rewards, certificates, badges, etc., through human–computer interaction and interpersonal interactions. The platforms offer special rewards that are novel and arousing to users, which could be perceived as an unusual experience and generate higher self-expansion [51]. Ant Forest is a good example of providing such new experiences. On Ant Forest, “green energy” can be accumulated by users through low-carbon behavior, such as green purchases and public transportation, and when the “green energy” reaches the required amount, Alibaba will donate a tree and plant it in desert areas. Consumers may take pleasure in several interesting activities on Ant Forest, including collecting green energy, “stealing” energy, and planting trees with their friends. In addition, Ant Forest constantly introduces gameplay innovation, brings in new tree species, and updates the interface and content of the game, maintaining the novelty of it. The gamification design with multiple sensory modalities and challenging game rules allows for relaxation and self-immersion. The platforms can play a powerful role when they continuously provide pleasant experiences through aesthetic or hedonic elements with immediate mood-altering properties [93]. Such platforms trigger users’ self-expansion, and affect people’s emotions, such as hope, efficacy, ability to cope with life problems, optimism towards daily distress management, and emotional stability [94].

Competence-based expansion refers to raising one’s capability. People can achieve self-expansion by participating in novel and challenging activities [18,19]. VCSRC might be such an activity. Consumers achieve their goals (such as planting a tree in Ant Forest, giving help to the people who are in need, and achieving a certain rank) and obtain positive feedback in the co-creation activities, which can improve self-esteem and sense of capability, and trigger self-expansion [51]. The platforms also present abundant knowledge containing various cognitive and affective components through rich media, dealing with subjects related to nature protection, caring for disadvantaged groups, rural revitalization, etc. When immersed in the platforms, consumers enrich their knowledge, increase social experience, and perceive the positive energy of society, thus having the feeling of extending their possibilities. On a deeper level, one might feel empathy or hope, involving sensations such as awe, elevation, and flow (the sense of being completely immersed in something [95]), and giving rise to an increased appreciation for moral virtue which correlates with thoughts and feelings directed beyond the self [96].

Identity-based expansion refers to obtaining new identities and roles. Self-expansion theory suggests that individuals are intrinsically driven to seek opportunities to cultivate new perspectives and obtain new identities [17]. They perceive a brand from various aspects, connected to their actual self or ideal self [97], or with cultural ideals that are widespread in society [98]. Brands reflecting ideal values can trigger consumers’ self-expansion [22]. VCSRC platforms are such “brands” that symbolize ideal values (personal ideal or cultural ideal) that might provide new identities and roles that enable consumers to expand their self-identity in a desirable direction. For example, in the turbulent situation of the COVID-19 crisis, the values of benevolence and altruism seem to be a new ideal, prompting people to share and donate excessive drugs via platforms, helping each other bridge over the difficulties. The VCSRC platforms give consumers new identities, such as “Love Ambassador”, “Nature Warden”, etc. The role that consumers play on the platforms might be the ideal self in their minds who they desire and strive to be [22]. In addition, people can make new friends on the platforms, thus new identities and roles will be given, which can also bring opportunities for self-expansion [99].

Self-expansion theory indicates that if a brand enables consumers to do something special and exciting, they are likely to have a more positive attitude towards the brand [51]. Evidence suggested that the self-expansion experience is rewarding, and this positive affection drives sustained maintenance and development of the relationship [24,100]. When membership provides significant benefits, people with high self-expansion are greatly motivated to join and maintain their member status [101]. That is, if an individual believes that the current relationship will provide self-expansion in the future, there is almost no reason to leave the relationship or search for alternatives [56]. Moreover, self- expansion is psychologically healthy, leading to a broadened identity or awareness, which usually results in greater altruism [24]. According to the above discussions, we proposed the following:

 **Hypothesis 1a (H1a).**
*Experience-based expansion has a positive effect on consumers’ continuous participation in VCSRC.*


 **Hypothesis 1b (H1b).**
*Competence-based expansion has a positive effect on consumers’ continuous participation in VCSRC.*


 **Hypothesis 1c (H1c).**
*Identity-based expansion has a positive effect on consumers’ continuous participation in VCSRC.*


### 3.2. Self-Expansion and Consumer Engagement

The positive effects of self-expansion were demonstrated in several studies. For example, Michel et al. [22] proved that self-expansion leads to consumers’ favorable responses to brands, including brand attitude, purchase intentions, and recommendation intention. VCSRC platforms, similar to brands, can offer consumers plenty of resources that facilitate self-expansion, resulting in self-efficacy and psychological richness [102], which might induce positive results such as consumer engagement.

First of all, people have a fundamental motive to seek self-expansive opportunities such as enjoyable experiences [19]. The motivation of pleasure seeking drives consumers to engage with brands [103]. If consumers obtain pleasure and enjoyment in a virtual community, they are more inclined to devote time or effort to it [104]. Empirical findings also show that users are more passionate about participating in online activities where they receive more pleasure [79]. Carlson et al. [105] proved that the hedonic value brought by the brand community will lead to consumers’ stronger willingness to feedback to the brand and cooperate with others in the community.

Secondly, by expanding the self, people feel more positively about the self and have more resources upon which to draw, thereby enhancing their sense of self-efficacy [19,53,54]. Self-efficacy can determine a person’s behavior, effort level, and emotional response mode, affecting the person’s degree of involvement in activities and the person’s endurance to continue activities when facing difficulties, obstacles, setbacks, and failures [106]. Khan [107] put forward the concept of social media self-efficacy and demonstrated that it is significantly related to CE. The integration of accumulated value, ranking, medals, and other game elements into virtual CSR activities can effectively manage the process of consumer participation, which brings about a continuous positive feedback mechanism that is also conducive to enhancing consumers’ self-efficacy and optimizing consumer experience [108]. The formation of this sense of self-efficacy will produce the role of self-motivation and promote the formation of CE [92]. Wang et al. [109] proved that users with high self-efficacy will more actively share knowledge and establish long-term relationships with other users with the same interests and needs. Consumers with a high sense of self-expansion believe in their capabilities to complete the participation behavior and that their participation is highly likely to succeed. A strong sense of self-expansion encourages a person to try a highly challenging job, set a high level of goals, collect, integrate, and even think about brand-related information, have more positive feelings, and show a strong goal commitment, so that compared with other platforms of the same kind, they are willing to pay more for this specific platform, such as spending more time browsing or giving priority [110]. As explained by Park et al. [56], strong attachments will arise when a brand brings a sense of a capable and efficacious self.

Furthermore, psychological richness received attention as another dimension of well-being recently. According to Oishi and Westgate [102], experiences that are novel, challenging, and perspective-changing lead to psychological richness. This effect is driven by the feeling of self-expansion. People high in psychological richness report greater curiosity and tend to seek out challenges and opportunities for learning [102]. Rodas [111] further found that brands with psychological richness due to self-expansion led to higher consumer evaluations, which in turn led to positive brand perception. Consumers’ self-identification with brands will significantly strengthen their engagement and word-of-mouth recommendation intentions [112]. According to the above discussions, we proposed the following:

 **Hypothesis 2 (H2).**
*Experience-based expansion has a positive effect on (a) conscious attention, (b) enthusiasm, and (c) social connection.*


 **Hypothesis 3 (H3).**
*Competence-based expansion has a positive effect on (a) conscious attention, (b) enthusiasm, and (c) social connection.*


 **Hypothesis 4 (H4).**
*Identity-based expansion has a positive effect on (a) conscious attention, (b) enthusiasm, and (c) social connection.*


### 3.3. Consumer Engagement and Continuous Participation

Previous studies showed that brand loyalty is one of the major outcomes of CE [65]. In addition, attachment, satisfaction, trust, and self–brand connection may all be outcomes of CE [64,65,113]. Many studies proved that these factors have positive impacts on users’ continuous participation behavior [114,115]. Therefore, we can suppose that CE will have a positive effect on users’ continuous participation, which also received some empirical support (e.g., [37,68,113,116]). The literature also suggests that when there is a strong attachment to others, people are more inclined to make sacrifices and private investments to support the continuation of the relationship [117]. In a marketing context, consumers may devote their resources, such as time, money, and energy, to maintain their relationship with brands [93].

In this study, conscious attention refers to the degree of consumers’ desire or interest in the VCSRC platforms. Consumers who are highly interested in a certain social media platform will be more familiar with the products and services of the platform enterprises, more easily understand the characteristics of the platform, and be more likely to keep using or recommend the platform to others [118]. Secondly, enthusiasm is the consumers’ emotional and enthusiastic response to the platforms. The enthusiastically participating consumers will actively take part in the activities of the platforms, and will more actively spread word of mouth for the platforms. These behaviors will strengthen the relationships between the platforms and consumers and enhance consumer loyalty. Jaakkola and Alexander [44] suggested that consumers who actively participate in enterprise activities typically show stronger brand loyalty. Thirdly, social connection is the enhancement of the interactions with others related to the platforms. Its connotation includes the reciprocal relationship between members, bringing benefits to each other. Consumers can actively interact with the social media platform or devote resources, which may lead to contributing product knowledge and use skills to help each other, and may also bring pleasant experiences. In the process of mutual help, consumers with common interests, goals, and needs may be found to establish a long-term relationship. Through such a mutually beneficial interaction process, consumers will strengthen loyalty attitudes and behaviors. Past research showed that interaction is a significant experience in the process of using media [119]. Individuals pursue social satisfaction from the interactions that promote their subsequent usage behavior [120]. Therefore, CE is likely to have a positive effect on continuous participation in VCSRC. According to the above discussions, we proposed the following:

 **Hypothesis 5a (H5a).**
*Conscious attention has a positive effect on consumers’ continuous participation in VCSRC.*


 **Hypothesis 5b (H5b).**
*Enthusiasm has a positive effect on consumers’ continuous participation in VCSRC.*


 **Hypothesis 5c (H5c).**
*Social connection has a positive effect on consumers’ continuous participation in VCSRC.*


Based on theoretical hypotheses, a conceptual model of the relationship among consumers’ self-expansion, consumer engagement, and continuous participation in VCSRC was constructed (see Figure 1).

## 4. Methods

This study aimed to explore the driving factors of consumers’ continuous participation in VCSRC practices. By integrating self-expansion and CE theory, a theoretical model was constructed. We conducted a survey study through a series of steps, including questionnaire design, questionnaire distribution, data collection, and data analysis. Considering that Ant Forest is the most popular VCSRC platform in China with 650 million users, and that the relatively wide range of user groups is conducive to the collection of questionnaire data, we select Ant Forest as the research scenario in this study, providing a guarantee for the credibility and validity of the study. Based on the questionnaire data of 336 Ant Forest users, the hypotheses were tested by SEM.

### 4.1. Data Collection

A questionnaire was used for data collection. Drawing on existing literature and scales, a survey questionnaire was designed for this study. We created the questionnaire on “Questionnaire Star,” the largest online questionnaire survey site in China. Two professors and eight Ph.D. students who had experience with Ant Forest were invited to participate in a pilot test to ensure that the response time and wording of the questionnaire were appropriate. Then, we distributed the questionnaire via social media platforms such as QQ and WeChat, which are the most popular instant messengers in China, to invite target users to participate. The survey objects were individual users who were continuously using Ant Forest for at least six months. The questionnaire survey was conducted from June to August 2022, lasting about 3 months. A total of 435 questionnaires were distributed and 380 were returned, a recovery rate of 87.4%. Finally, 336 valid questionnaires were recovered. The majority of respondents were young people aged 18–30, who are the dominant users of Ant Forest. The sample details are shown in Table 2.

### 4.2. Measures

All variables were measured with a seven-point Likert scale. After a small-sample test and question item purification, the final scale was formed. Self-expansion mainly refers to the research of Lewandowski and Aron [121] and Niu et al. [27]. The 14-item Self-Expansion Questionnaire (SEQ), developed by Lewandowski and Aron [121], is a widely used measure of relational self-expansion, and it was adapted by scholars to measure individuals’ self-expansion in different fields. Sample items include “How many novel experiences has your partner provided you?”, “To what extent do you think your partner has helped you become a better person?”, and “ How much does your partner increase your ability to accomplish new things?” The question format was converted into a declarative sentence in our scale. We also referred to the research of Niu et al. [27] who identified three dimensions of self-expansion with 16 items to assess internet self-expansion. Drawing on the work of Lewandowski and Aron [121] and Niu et al. [27], we divided self-expansion into three dimensions and adapted a 13-item scale so that it would be relevant to a VCSRC context. CE was measured from three dimensions with a total of 10 items, adapted from the measurement methods of Vivek et al. [70] and Zhang et al. [37]; continuous participation was adapted from Osei-Frimpong et al. [5], with a total of 3 items.

### 4.3. Data Analysis

We followed the two-step approach recommended by Anderson and Gerbing [122] to investigate the proposed model. First, we checked the measurement model to evaluate the reliability and validity of the scale, using exploratory factor analysis (EFA) and confirmatory factor analysis (CFA). Second, we used the structural equation model (SEM) to test the hypotheses. The SPSS26.0 and AMOS23.0 software packages were adopted for testing.

## 5. Results

### 5.1. Reliability and Validity Test

First of all, SPSS26.0 was used for KMO and Bartlett testing. We can see from Table 3 that the KMO value was 0.894, which is higher than 0.7. For this reason, the questionnaire data met the conditions for factor analysis.

Secondly, SPSS26.0 was used for the exploratory factor analysis. Table 4 shows that combined with the factor loadings, 26 items in the questionnaire were classified into seven factors. The factor dimensions obtained from the rotated factor loading matrix are consistent with the research expectations, and the absolute value of the factor loadings were higher than 0.5 [123]. Therefore, the questionnaire in this paper has good structural validity.

Thirdly, we used SPSS26.0 software to conduct the reliability test and AMOS23.0 software for the confirmatory factor analysis; the results are summarized in Table 5. The Cronbach’s α values of all variables were higher than 0.7, indicating that the scale has good internal consistency according to Nunnally [124] and Churchill [125]. All items loaded on the expected factors were higher than 0.7, the t values were all higher than 2.0, the composite reliability (CR) values were all higher than 0.8, and the average variance extracted (AVE) values were higher than 0.5. The values of each fitting index in the confirmatory factor analysis of this study (χ^2^ = 448.883, df = 278, χ^2^/df = 1.615 < 3, NFI = 0.911 > 0.9, IFI = 0.964 > 0.9, TLI = 0.958 > 0.9, CFI = 0.964 > 0.9, GFI = 0.875 > 0.8, and RMSEA = 0.051 < 0.08) all met the recommended value or were in an acceptable range, demonstrating that the confirmatory factor analysis model had a good fitting effect on the sample data obtained. It can be seen that the scale used in our research presents good reliability and convergent validity, and the theoretical model and research hypotheses are reasonable, so they can be used for further analysis of the relationships among the various variables.

Finally, SPSS26.0 was used to perform Pearson correlation analysis. Table 6 shows that the square root of AVE on the diagonal was greater than the correlation coefficient in the corresponding row, indicating that the scale had good discriminant validity.

### 5.2. Autocorrelation and Multicollinearity

Taking experience-based expansion, competence-based expansion, identity-based expansion, conscious attention, enthusiasm, and social connection as independent variables, and continuous participation as the dependent variable for the linear regression analysis, the results show that the adjusted R-squared value of the model was 0.453, which means the independent variables can explain 45.3% of the changes in the dependent variable. The Durbin–Watson value was 2.293, which is close to 2, indicating that the sample data had no autocorrelation [127].

To test whether there was multicollinearity in our data, the variance inflation factor (VIF) was estimated. The results show that the VIF values ranged from 1.208 to 1.542, which were less than the threshold of 5, indicating that there was no obvious multicollinearity among the variables [127].

### 5.3. Common Method Variance

To test for common method bias, we used Harman’s [128] single factor test, which performs an exploratory factor analysis (EFA) of all measurement items. The results show that the variance explained by the first factor was 36.493%, which is under the critical value of 50%, indicating that the common method bias of the data is acceptable [129].

### 5.4. Structural Equation Model Analysis

AMOS23.0 was adopted to conduct structural equation model analysis and hypothesis tests. The output of the SEM and analysis results are presented in Figure 2 and Table 7. It can be seen from the table that the fit indices of the SEM (χ^2^ = 656.167, df = 281, χ^2^/df = 2.335 < 3, NFI = 0.903 > 0.9, IFI = 0.942 > 0.9, TLI = 0.933 > 0.9, CFI = 0.942 > 0.9, RMSEA = 0.063 < 0.08, and GFI = 0.875, close to 0.9) all met the recommended standards or were within the acceptable range [103,104]. The standardized path coefficients between all the explicit and latent variables were higher than 0.5, and the corresponding C.R. values were higher than the critical value of 1.96, and at least at the *p* = 0.05 level, were statistically significant. The path C.R. values between all endogenous and exogenous latent variables were higher than 1.96, and at least at the *p* = 0.05 level, were statistically significant. This shows that the SEM fit well. All assumptions were effectively verified.

### 5.5. Assessment of Mediating Effects of CE

We also tested the mediating effects of the three dimensions of CE by using the bootstrap method of Amos 23.0. The number of bootstrap samples was set to 5000, and the percentile confidence interval was set to 95%. When the bias-corrected confidence intervals of indirect effect do not overlap with zero, it indicates that a mediating effect exists [126]. As shown in Table 8, for the three dimensions of self-expansion, all the confidence intervals of the indirect effects did not overlap with zero. For example, for the indirect effect from EBE to CP via CA, the effect value was 0.043, the lower bound of the bias-corrected confidence interval was 0.009, and the up bound was 0.100. Therefore, the mediating effects of the three dimensions of CE in the paths exist. These findings further confirm our previous tests. Moreover, all the confidence intervals of the direct effects did not overlap with zero, indicating that the three dimensions of CE partially mediate the impact of the three dimensions of self-expansion on CP.

## 6. Conclusions and Discussion

In this study, we conducted a survey to investigate consumers’ continuous participation in VCSRC by examining the roles of self-expansion and consumer engagement in this process. The main findings of our research are as follows:

Not limited to hedonic or utilitarian perspectives, which the existing research of continuance intention generally focused on (e.g., [12,49,130,131]), this study revealed the self-expansion motivation of consumers’ continuous participation in VCSRC, such as to broaden one’s perspective on things, to discover oneself, and to be a better person, which is similar to the research of Ashfaq et al. [29] and Zhang et al. [16] on self-transcendence and self-promotion motivations. The results show that the three dimensions of self-expansion proposed in this study, including experience-based expansion, competence-based expansion, and identity-based expansion, all have significant positive impacts on continuous participation in VCSRC. These findings extend the research of Ashfaq et al. [29] and Zhang et al. [16], and support Liu et al. [28], who found a positive relationship between self-expansion and loyalty in non-relationship contexts, which is rarely studied in the existing literature. Furthermore, we divided self-expansion into three dimensions instead of treating it as a single dimensional concept, thus deepening the research of Liu et al. [28].

A few studies identified self-expansion as an antecedent of CE, while this study makes an attempt to confirm this relationship. The results indicate that the three dimensions of self-expansion all have significant positive effects on the dimensions of CE, including conscious attention, enthusiasm, and social connection. These results further confirm the findings of Mattingly and Lewandowski [56], who suggested that individuals who experienced greater self-expansion exerted more effort on cognitive and physical tasks. These results also support and extend the existing research on the motivational factors for CE by Huang et al. [79], Parihar et al. [103], Luo et al. [104], Bazi et al. [88], Khan et al. [107], and Wang et al. [109].

The three dimensions of CE were all positively related to continuous participation in VCSRC, indicating that engaged consumers are likely to continue participating. These findings support the research of [32,33,37], who also found a positive relationship between CE and continuance, whereas CE acting as a partial mediator between self-expansion and continuous participation in VCSRC is a new finding that we added to the literature. Therefore, the positive effect of self-expansion on continuous participation in VCSRC is partially achieved through CE, extending the findings of Yang et al. [12] and Ashfaq et al. [29].

### 6.1. Theoretical Implications

This study made several theoretical contributions. First of all, we made a significant attempt to explain why consumers continuously participate in VCSRC from a novel perspective of self-expansion, examining the influence mechanism of self-expansion on continuous participation via CE. The findings highlight self-expansion and CE as factors driving this continuous practice. CE represents an individual’s psychological state of engagement with VCSRC platforms, which leads to continuous participation in such practices, whereas self-expansion expounds the motives in driving consumers’ continuous participation. Hence, we integrate the driving factors from self-expansion and CE to conceptualize continuous VCSRC practices. Given the limited attention in the literature, our research responds to the calls for further study into VCSRC practices [11] and provides valuable insights into consumers’ continuous participation in such practices.

Second, this study extends the concept of self-expansion to the context of VCSRC, enriching the research on self-expansion. Liu et al. [28] suggested that non-relational self-expansion merits more research, whereas it is largely unexplored in the existing literature. Hence, this study contributes to the limited literature by investigating the connotation and positive effects of self-expansion in VCSRC practices. In this vein, our research deepens the self-expansion theory and contributes to the literature on the role of self-expansion motives in consumer behavior.

Third, the concept of CE, a hot topic in the marketing field, is introduced into the VCSRC situation. Combining with the reality of the VCSRC practices, the connotation, dimensions, antecedents, and consequences of CE were explored, thus enriching and expanding the research on CE.

### 6.2. Practical Implications

This study provides a reference for the managers of VCSRC projects to promote consumers’ continuous participation. The first implication of this study is that the managers should focus on satisfying users’ self-expansion motivations to engage their consumers. Specifically, when designing the platforms, managers could consider adopting gamification principles and updating the aesthetic design and functionality of the platform regularly to bring new experiences to users. They should also bring forth new ideas to provide users with special rewards and feedbacks, such as the positive effects of their actions on the real world, enabling them to perceive the results of their efforts in more ways to stimulate their sense of ability. Meaningful information and knowledge could be presented on the platforms with rich media to increase users’ knowledge and open up their perspectives. Additionally, managers would be well advised to help users expand their sense of the self; for instance, incorporating into the VCSRC practice symbolic meanings, such as endowing the users with new identities and embodying ideal values on the platforms, to prompt consumers to expand their vision and to reveal new aspects of the self. Another implication of this study is that the operators of the virtual CSR co-creation project should focus on strategies to cultivate the users’ engagement in the project. For example, the platforms should be more interesting and interactive in order to win consumers’ attention and increase their enthusiasm and interaction.

### 6.3. Limitations and Future Research

There are some limitations that need to be considered in our study. First, this study mainly focused on factors that could drive consumers’ continuous participation in VCSRC at the individual level. Other potentially important factors, such as social and situational factors that could also drive such behaviors, were not taken into account. Future studies could extend on our study by exploring the influences of these factors. It will also be interesting to consider the moderating factors, such as participation frequency and consumers’ personality traits, to investigate potential changes in the dynamics of continuous VCSRC practices.

Second, this study defined continuous participation in VCSRC as having followed it for at least six months; future research could adopt a longitudinal method and investigate such behaviors at different time intervals.

Third, the empirical study only focused on Ant Forest in China, which might be a limitation. Although the most representative case was selected, the conclusions drawn need to be extended to a wider range of VCSRC platforms, and comparative studies for different types of platforms should be conducted to determine any potential differences.

## Figures and Tables

**Figure 1 behavsci-13-00545-f001:**
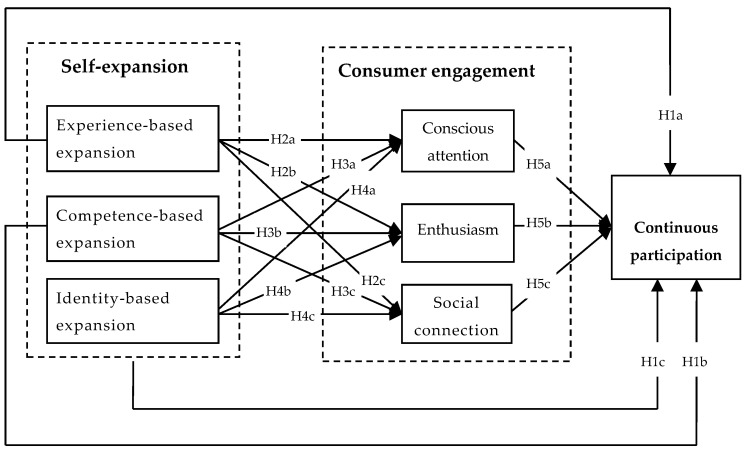
The conceptual model of the relationship among consumers’ self-expansion, consumer engagement, and continuous participation in VCSRC.

**Figure 2 behavsci-13-00545-f002:**
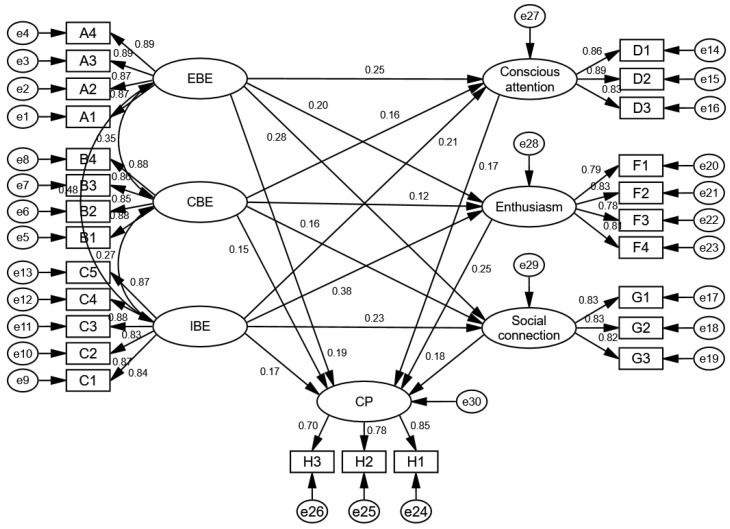
The output of the SEM analysis. Note: EBE = experience-based expansion, CBE = competence-based expansion; IBE = identity-based expansion; and CP = continuous participation.

**Table 2 behavsci-13-00545-t002:** Sample descriptive statistics.

Item	Category	Frequency	Percentage (%)
Gender	Male	138	41.07
	Female	198	58.93
Age	18–25	164	48.81
	26–30	95	28.27
	31–40	55	16.37
	41–50	20	5.95
	51–60	2	0.60
Education	Junior college or below	5	1.49
	Undergraduate course	263	78.27
	Master’s degree	56	16.67
	Doctorate and above	12	3.57
Monthly income	Less than CNY 2000	131	38.99
	CNY 2000–5000	55	16.37
	CNY 5000–8000	89	26.49
	CNY 8000–10,000	42	12.50
	More than CNY 10,000	19	5.65
Time using Ant Forest	6 months–1 year	28	8.33
	1–2 years	41	12.20
	2–3 years	65	19.35
	3 years or more	202	60.12
Frequency of using Ant Forest	Once a day	86	25.60
	Several times a day	45	13.39
	2–3 times a week	167	49.70
	Several times a month	31	9.23
	Once a month	7	2.08
Total		336	100

**Table 3 behavsci-13-00545-t003:** Results of KMO and Bartlett testing.

KMO and Bartlett Testing		
KMO value		0.894
Bartlett’s test of sphericity	Approximate chi-square	4860.458
	Freedom	325
	Significance	0.000

**Table 4 behavsci-13-00545-t004:** Results of exploratory factor analysis.

Rotated Component Matrix							
	Factor loadings						
	Factor 1	Factor 2	Factor 3	Factor 4	Factor 5	Factor 6	Factor 7
EBE 1	0.209	0.169	0.844	0.156	0.136	0.073	0.120
EBE 2	0.235	0.111	0.803	0.218	0.168	0.167	0.138
EBE 3	0.210	0.159	0.847	0.124	0.116	0.156	0.172
EBE 4	0.219	0.119	0.836	0.146	0.136	0.182	0.139
CBE 1	0.045	0.897	0.099	0.114	0.105	0.041	0.105
CBE 2	0.108	0.870	0.114	0.076	0.111	0.072	0.099
CBE 3	0.101	0.878	0.154	0.052	0.135	0.064	0.095
CBE 4	0.080	0.878	0.101	0.114	0.044	0.125	0.125
IBE 1	0.833	0.120	0.185	0.205	0.030	0.116	0.105
IBE 2	0.851	0.089	0.155	0.197	0.087	0.116	0.097
IBE 3	0.797	0.039	0.195	0.181	0.178	0.128	0.115
IBE 4	0.803	0.120	0.155	0.210	0.148	0.152	0.203
IBE 5	0.799	0.056	0.236	0.165	0.171	0.082	0.224
CA 1	0.145	0.154	0.133	0.085	0.857	0.093	0.143
CA 2	0.121	0.081	0.138	0.045	0.887	0.084	0.158
CA 3	0.167	0.137	0.156	0.067	0.859	0.044	0.046
ENT 1	0.203	0.083	0.045	0.852	0.045	0.049	0.095
ENT 2	0.192	0.091	0.170	0.812	0.023	0.114	0.137
ENT 3	0.260	0.100	0.232	0.737	0.042	0.086	0.187
ENT 4	0.165	0.100	0.146	0.845	0.116	−0.016	0.102
SC1	0.154	0.077	0.159	0.118	0.073	0.832	0.165
SC2	0.145	0.023	0.153	0.039	0.070	0.850	0.193
SC3	0.131	0.176	0.126	0.035	0.075	0.865	0.035
CP 1	0.268	0.177	0.221	0.247	0.116	0.176	0.735
CP 2	0.222	0.208	0.229	0.107	0.200	0.131	0.760
CP 3	0.216	0.144	0.134	0.262	0.137	0.215	0.703

Note: EBE = experience-based expansion, CBE = competence-based expansion; IBE = identity-based expansion; CA = conscious attention; ENT = enthusiasm; SC = social connection; and CP = continuous participation.

**Table 5 behavsci-13-00545-t005:** Results of reliability test and confirmatory factor analysis of variables.

Constructs	Measurement Items						Factor Loading	t-Value	CR	AVE
EBE (α = 0.940)	EBE1: Ant Forest provides me with new experiences.						0.882	-	0.940	0.797
	EBE2: Ant Forest provides me with exciting experiences.						0.879	19.085		
	EBE3: Ant Forest provides me with a larger perspective on things.						0.909	20.491		
	EBE4: Ant Forest expands my understanding of external things.						0.901	20.070		
CBE (α = 0.933)	CBE1: Ant Forest increases my ability to accomplish new things.						0.897	-	0.933	0.776
	CBE2: I have learned new things through Ant Forest.						0.863	18.668		
	CBE3: Ant Forest increases my knowledge.						0.882	19.524		
	CBE4: With Ant Forest, I have the feeling of extending my possibilities.						0.882	19.523		
IBE (α = 0.936)	IBE1: I have different identities and roles in Ant Forest, which are different from those in real life.						0.857	-	0.937	0.747
	IBE2: I find different aspects of myself in Ant Forest.						0.870	17.548		
	IBE3: Ant Forest helps me to expand my sense of the kind of person I am.						0.838	16.406		
	IBE4: Ant Forest makes me rediscover myself.						0.882	17.985		
	IBE5: Being a member of Ant Forest has made me a better person.						0.874	17.701		
CA (α = 0.901)	CA1: Anything related to Ant Forest grasps my attention.						0.873	-	0.901	0.753
	CA2: I like to learn more about Ant Forest.						0.900	17.447		
	CA3: I pay a lot of attention to anything about Ant Forest.						0.829	15.771		
ENT (α = 0.893)	ENT1: I spend a lot of my discretionary time on Ant Forest.						0.834	14.461		
	ENT2: I am passionate about Ant Forest.						0.839	-	0.874	0.698
	ENT3: I am heavily into Ant Forest.						0.845	14.185		
	ENT4: My days would not be the same without Ant Forest.						0.823	13.853		
SC (α = 0.873)	SC1: I love participating in Ant Forest with my friends.						0.824	-	0.893	0.677
	SC2: I enjoy taking part in Ant Forest more when I am with others.						0.828	14.321		
	SC3: Participating in Ant Forest is more fun when other people around me do it too.						0.804	13.763		
CP(α = 0.842)	CP1: I continuously use Ant Forest.						0.864	-	0.846	0.647
	CP2: I continuously participate in the activities on Ant Forest.						0.811	13.953		
	CP3: I continuously participate in Ant Forest to enable me to reach personal goals.						0.733	12.277		
Fit indices	χ^2^	df	χ^2^/df	NFI	IFI	TLI	CFI	GFI	RMSEA	
Fit of the model	448.883	278	1.615	0.911	0.964	0.958	0.964	0.875	0.051	
Recommended value (Wu, 2009) [126]	/	/	<3	>0.9	>0.9	>0.9	>0.9	>0.9	<0.080	

Note: EBE = experience-based expansion, CBE = competence-based expansion; IBE = identity-based expansion; CA = conscious attention; ENT = enthusiasm; SC = social connection; and CP = continuous participation.

**Table 6 behavsci-13-00545-t006:** Square root of AVE and Pearson correlation analysis.

	EBE	CBE	IBE	CA	ENT	SC	CP
EBE	0.893						
CBE	0.349 **	0.881					
IBE	0.524 **	0.269 **	0.864				
CA	0.387 **	0.296 **	0.367 **	0.868			
ENT	0.431 **	0.268 **	0.504 **	0.231 **	0.835		
SC	0.397 **	0.245 **	0.369 **	0.241 **	0.234 **	0.823	
CP	0.525 **	0.404 **	0.554 **	0.410 **	0.492 **	0.445 **	0.804

Note: EBE = experience-based expansion, CBE = competence-based expansion; IBE = identity-based expansion; CA = conscious attention; ENT = enthusiasm; SC = social connection; and CP = continuous participation. ** *p* < 0.01.

**Table 7 behavsci-13-00545-t007:** Analysis results of the SEM.

Path			Estimates	S.E.		C.R.	*p*		Standardization Coefficient		Result
EBE	--->	CA	0.248	0.064		3.881	***		0.255		Supported
CBE	--->	CA	0.157	0.058		2.719	0.007		0.159		Supported
IBE	--->	CA	0.202	0.062		3.244	0.001		0.206		Supported
EBE	--->	ENT	0.164	0.053		3.109	0.002		0.198		Supported
CBE	--->	ENT	0.098	0.047		2.053	0.040		0.117		Supported
IBE	--->	ENT	0.315	0.054		5.884	***		0.378		Supported
IBE	--->	SC	0.187	0.052		3.585	***		0.229		Supported
CBE	--->	SC	0.134	0.048		2.772	0.006		0.163		Supported
EBE	--->	SC	0.224	0.054		4.178	***		0.276		Supported
EBE	--->	CP	0.168	0.053		3.150	0.002		0.191		Supported
CBE	--->	CP	0.132	0.046		2.867	0.004		0.149		Supported
IBE	--->	CP	0.150	0.054		2.771	0.006		0.170		Supported
CA	--->	CP	0.150	0.049		3.082	0.002		0.167		Supported
ENT	--->	CP	0.264	0.063		4.188	***		0.249		Supported
SC	--->	CP	0.196	0.062		3.154	0.002		0.181		Supported
Fit indices			χ^2^	df	χ^2^/df	NFI	IFI	TLI	CFI	GFI	RMSEA
Fit of the model			656.167	281	2.335	0.903	0.942	0.933	0.942	0.875	0.063
Recommended value (Wu, 2009) [126]			/	/	<3	>0.9	>0.9	>0.9	>0.9	>0.9	<0.080

Note: EBE = experience-based expansion, CBE = competence-based expansion; IBE = identity-based expansion; CA = conscious attention; ENT = enthusiasm; SC = social connection; and CP = continuous participation. *** *p* < 0.001.

**Table 8 behavsci-13-00545-t008:** Results of mediating effect testing.

Path	Effect Values	BootSE	BootLLCI	BootULCI
Total effects				
EBE ---> CP	0.333	0.070	0.197	0.468
CBE ---> CP	0.234	0.059	0.114	0.345
IBE ---> CP	0.340	0.073	0.196	0.481
Direct effects				
EBE ---> CP	0.191	0.076	0.046	0.342
CBE ---> CP	0.149	0.055	0.035	0.251
IBE ---> CP	0.170	0.076	0.018	0.317
Indirect effects				
EBE ---> CA ---> CP	0.043	0.022	0.009	0.100
EBE ---> ENT ---> CP	0.049	0.021	0.017	0.099
EBE ---> SC ---> CP	0.050	0.021	0.017	0.102
CBE ---> CA ---> CP	0.027	0.016	0.004	0.070
CBE ---> ENT --> CP	0.029	0.017	0.002	0.069
CBE ---> SC ---> CP	0.030	0.018	0.005	0.076
IBE ---> CA ---> CP	0.034	0.021	0.005	0.091
IBE ---> ENT ---> CP	0.094	0.029	0.049	0.165
IBE ---> SC ---> CP	0.041	0.022	0.009	0.104

Note: EBE = experience-based expansion, CBE = competence-based expansion; IBE = identity-based expansion; CA = conscious attention; ENT = enthusiasm; SC = social connection; and CP = continuous participation.

## Data Availability

The datasets generated and/or analyzed for the current study are available from the corresponding author on reasonable request.

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
