# Peer review of "The Influence of Self-Expansion and Consumer Engagement on Consumers’ Continuous Participation in Virtual Corporate Social Responsibility Co-Creation"

_behavsci, 2023, doi:10.3390/bs13070545_

Round 1
Reviewer 1 Report
Dear authors, your article is remarkable in terms of methodological coherence and bibliographic review. We appreciate the hypothesis design and the good balance between the practical and theoretical background.
Reviewer 2 Report
I have several concerns regarding this study.
-As self-expansion is one of the most important factors, and self-expansion-related studies should be described in detail with their studies, methods, and findings in a table form in the literature review section
-Have you considered other factors than self-expansion and consumer engagement?
-Most of the studies regarding the continuous intention to use or participate are mainly based on the TAM or ECM models. Any reason to choose the model in your study? Please clarify it.
Presentation and Style
Several issues need to be corrected:
1. On Page 10, the factor analysis results should be mentioned with the reference-the absolute value of the factor loadings are beyond 0.5. Therefore...
The reference regarding the standard and validity should be mentioned
2. What does CSR stand for? Do not assume that everyone knows the terms.
Use the full name the first time instead of the abbreviation.
3. As important information about the method is missing (e.g. the date or duration regarding the data collection, when it started and ended, how long)
4. As Self-expansion were referred from the research of Lewandoski and Aron,
there is a strong need that you should mention what are their measurement and how did you adjust their measurement for this manuscript.
5. At the beginning of the conclusion and discussion, the summarization part sound like repetitive. You need to delete or move it into other sections.
Reviewer 3 Report
Thank you for submitting your research output to Behavioral Sciences. The research work is quite interesting, and the authors' efforts are appreciable. Most of the scientific hallmarks of writing a research paper have been met. However, I suggest some additions to enhance its chance of publication in Behavioral Sciences:
1. The title needs to be rephrased. "what drives" is usually used to reflect exploratory studies when causes or predictive variables are unknown. You have already mentioned self-expansion and consumer engagement in the title as predictors.
2. The gap in the literature needs further elaboration. The introduction section's ending part presents a few lines to justify the present study but it is insufficient. You need to expand it and substantiate it with the latest literature.
3. The literature review part has been written quite nicely. However, the topic has a high currency and is being extensively investigated. The literature in the manuscript is relatively old and needs to be strengthened with the latest research work.
4. Your methodology should start with an explanation of the research design.
5. You should test to rule out the possibility of autocorrelation, multicollinearity and common method bias.
6. The discussion section is poorly and insufficiently written. It would be best to discuss hypothesis by hypothesis vis-à-vis literature and theoretical assumptions in detail. Mediations should also be elaborated in detail, explaining whether mediations were full or partial.
Goodluck
quality of English language is high. Minor corrections are required
Reviewer 4 Report
Dear authors, thanks for submission of the paper for publication. The paper aims at investigating the drivers of consumers’ continuous participation in virtual CSR co-creation from the perspective of self-expansion theory and by integrating consumer engagement. Based on the completed review the following strengths were identified as developed and supported methodology, appropriate analysis of data and interpretation of results and weaknesses as limited discussion of obtained results comparing to the results of previous similar studies, also hypotheses H1, H2 and H3 can be revised as they are not tested directly.
Thus, to improve the quality of this paper the following recommendations can be given as:
Figure 1. The conceptual model: please label each arrow (hypothesis) of the model.
Please revise your hypotheses, as H1, H2 and H3 were no tested directly. H1, H2 and H3 can be paraphrased as research questions which can be answered with few related hypotheses.
Discussion section should be expanded, please add some comparison of your obtained results with previous similar studies, identify the similarities and differences.
Reviewer 5 Report
The paper deals with a real and important topic, which is consistent with the purpose of the journal.
Overall the paper is nicely written, builds on appropriate theories and methods, and presents some good results. I believe the authors have done a good job in their work, and have managed to present a well-rounded manuscript.
Round 2
Reviewer 2 Report
Thank you for your input. I appreciate your response. I only have one concern regarding this(see below in blue font).
Response 2: Thank you for your insightful question. Yes, we considered that there could be other driving factors for consumers’ continuous participation in virtual CSR co-creation, such as consumer factors, virtual platform factors, enterprise factors, and environmental factors. Among these factors, consumer factors are particularly concerned, including consumers’ personality characteristics, values (e.g. altruism), intrinsic motivations (e.g. gratifications, perceived value), extrinsic motivations (e.g. rewards for activities), user experience (e.g. cognitive experience, affective experience), psychological states(e.g. satisfaction, attachment). Besides, virtual platform factors (e.g. gamification design elements, compatibility with lifestyle), enterprise factors( e.g. corporate reputaion), and environmental factors (e.g. environmental concern) may also deserve attention. These factors have been mentioned in the literature review part(See section 2.1 on page 3).
l Still unlclear how comes out two factors (self-expansion and consumer engangement) from all above driving factors for consumer continuous participation in virtual CSR co-creation. Please provide the detailed process or explation for this. Asking around experts? Is there any scientific way to prove that these two factors are worth to investigate?
Reviewer 3 Report
Thank you for submitting the revised version of the manuscript. You have done a good job. All the comments have been addressed skillfully. I am satisfied and the manuscript is much better than the previous version.
Need proofreading
Author Response
Thank you very much for your recognition of our revised version of the manuscript. We do appreciate your time and efforts on reviewing it. Your affirmation is the driving force for our progress.
Reviewer 4 Report
Dear authors, thanks for submission of revised paper, the quality of the paper improved, and it can be suggested for publication in present form.
Author Response

(The authors gave the same response as above.)
